


# Evaluation of Turbulence Measurement Techniques from a Single Doppler Lidar

Timothy A. Bonin[1,2], Aditya Choukulkar[1,2], W. Alan Brewer[2], Scott P. Sandberg[2], Ann M. Weickmann[1,2], Yelena Pichugina[1,2], Robert M. Banta[2], Steven P. Oncley[3], and Daniel E. Wolfe[4]

[1]Cooperative Institute for Research in Environmental Sciences, University of Colorado, Boulder, Colorado, USA
[2]Chemical Sciences Division, National Oceanic and Atmospheric Administration, Boulder, Colorado, USA
[3]National Center for Atmospheric Research, Boulder, Colorado, USA
[4]Physical Sciences Division, National Oceanic and Atmospheric Administration, Boulder, Colorado, USA

*Correspondence to:* T. A. Bonin (timothy.bonin@noaa.gov)

**Abstract.** Measurements of turbulence are essential to understand and quantify the transport and dispersal of heat, moisture, momentum, and trace gases within the planetary boundary layer. Through the years, various techniques to measure turbulence using Doppler lidar observations have been proposed. However, the accuracy of these measurements has rarely been validated against trusted *in situ* instrumentation. Herein, data from the eXperimental Planetary boundary layer Instrumentation Assess-

5 ment (XPIA) are used to verify Doppler lidar turbulence profiles through comparison with sonic anemometer measurements. For 17 days at the end of the experiment, a single scanning Doppler lidar continuously cycled through different turbulence measurement strategies: velocity azimuth display, six-beam, and range height indicators with a vertical stare.

Measurements of turbulence kinetic energy, turbulence intensity, and shear velocity from these techniques are compared with sonic anemometer measurements at six heights on a 300-m tower. The six-beam technique is found to generally measure
turbulence kinetic energy and turbulence intensity the most accurately at all heights, showing little bias in its observations. Turbulence measurements from the velocity azimuth display method tended to biased low near the surface, as large eddies were not captured by the scan. None of the methods evaluated were able to consistently accurately measure the shear velocity. Each of the scanning strategies assessed had its own strengths and limitations that need to be considered when selecting the method used in future experiments.

## 1 Introduction

Turbulence within the planetary boundary layer (PBL) transports and disperses heat, moisture, momentum, and other quantities. Additionally, atmospheric turbulence affects several disciplines and industries, such as wind energy, aviation, and air quality. For example, wind turbines may perform poorly and have a lower power output when turbulence intensity is large (Wharton and Lundquist, 2012; Choukulkar et al., 2016), and turbulence can shorten the lifespans of wind turbines (Kelley et al., 2006).

Pollutant dispersion from factories and other sources is primarily driven by advection and turbulent mixing within the PBL. Precise measurements are necessary to understand the role of turbulence within these disciplines, and to validate the turbulence





generated or parameterized in numerical weather prediction models and simulations. Scanning Doppler lidars are capable of addressing this need, by measuring vertical profiles of turbulent quantities throughout the entire PBL.

Turbulence has been measured using multiple Doppler lidars in recent years. Mann et al. (2009) performed the first intercomparison between triple-Doppler lidar turbulence measurements with sonic anemometry, and investigated the effects of volume averaging. Fuertes et al. (2014) used three Doppler lidars that were staring at the same point for 1-hr to retrieve a 0.5 Hz timeseries of $u$, $v$, and $w$. From these measurements, all six components of the Reynolds stress tensor ($\overline{u'^2}$, $\overline{v'^2}$, $\overline{w'^2}$, $\overline{u'w'}$, $\overline{v'w'}$, $\overline{u'v'}$) were directly calculated, showing good agreement with sonic anemometer observations. Newman et al. (2016a) used a similar method to measure velocity variances over several days, for comparison with other turbulence measurements. Additionally, Newman et al. (2016a) extended the use of 'virtual towers', which are constructed using measurements from multiple lidar systems, to measure vertical profiles of turbulence quantities, whereas in previous work 'virtual towers' were used to make profiles of the mean wind (e.g. Calhoun et al., 2006; Gunter et al., 2015). However, the long timeseries of staring data necessary to measure turbulence with meaningful significance (see Lenschow et al., 1994) precludes the possibility of making accurate turbulence profiles routinely, especially in the convective boundary layer where turbulent length scales are large. Doppler lidars are also expensive and many research groups do not have access to multiple systems. Hence, the focus in this paper will be on scanning strategies that provide vertical profiles of turbulence from a single Doppler lidar.

Over the years, several different techniques have been proposed to measure turbulence using measurements from a single Doppler lidar (Sathe et al., 2015a). A review of wind lidar turbulence measurements, through 2013, is provided by Sathe and Mann (2013). Since then, much of the recent work has focused on evaluating and improving turbulence measurements from the Doppler beam swinging (DBS) technique, which is often used by low-powered Doppler lidars typically used in the wind energy field (e.g., Lundquist et al., 2015; Newman et al., 2016b; Kumer et al., 2016). Herein, the DBS technique will not be evaluated since the exact methodology for the retrieval of turbulence statistics is still under development, with corrections being applied to reduce the effect of variance contamination and other error sources. Instead, three other scanning strategies were evaluated to determine their accuracies for measurement of turbulent quantities. The earliest proposed method to measure turbulence kinetic energy (TKE), individual velocity variances ($\overline{u'^2}$, $\overline{v'^2}$, $\overline{w'^2}$), and covariances ($\overline{u'w'}$, $\overline{v'w'}$, $\overline{u'v'}$) from Doppler lidar observations is discussed by Eberhard et al. (1989), using the residuals of a velocity-azimuth-display (VAD) fitting on a conical plan position indicator (PPI) scan. Banta et al. (2002, 2006) used vertical-slice range-height-indicator (RHI) scans to quantify the streamwise velocity variance $\overline{u'^2}$ within the PBL during nocturnal low-level jets. Vertical stares have been used in numerous studies (e.g. Pearson et al., 2010; Barlow et al., 2011; Bonin et al., 2015; Tonttila et al., 2015) to measure the boundary layer depth, vertical velocity spectra and variance, and dissipation. Recently, Sathe et al. (2015b) has proposed a method of using a six-beam strategy, which is essentially a modified DBS scan that typically uses five beams, to measure velocity variances, covariances, and TKE. Each of these strategies will be described in more detail within Sect. 2, and evaluated herein.

Turbulence measurements are needed to address a range of problems, which involve different breadths of the turbulence spectrum. All applications require accurate measurements of fluctuations by the largest energy-containing turbulent eddies and at least the lowest wavenumbers of the inertial subrange. Within the inertial subrange the magnitude of the fluctuations drops off





quickly (exponentially) with increasing wavenumber, so high wavenumbers make correspondingly smaller contributions to the total variances (Taylor, 1938). Detailed studies of turbulence dynamics, which may include studies of inflows to wind turbines or turbulence generated by them, may require accurate representation of fluctuations over the entire turbulence spectrum from large-eddy to dissipation scales (e.g., Troldborg and Sørensen, 2014). For such studies, employing the best data acquisition
strategies and understanding the errors involved is important. Other studies may not require this degree of precision. For example, evaluating the ability of NWP models to predict TKE involves values of 2–4 $\text{m}^2\text{ s}^{-2}$ in convective conditions and 1–2 $\text{m}^2\text{ s}^{-2}$ in weakly stable conditions. Such accuracies are achievable without measuring the entire spectrum. Many field programs are employing scanning lidar remote sensing in arrays to investigate spatial and temporal variations of the mean wind, as recommended in Banta et al. (2013). In such cases, the measurement of turbulence is not the primary goal, so the data-
acquisition and scanning approaches are not optimized for turbulence measurement. It is still desirable to obtain quantitative turbulence information (e.g., for NWP verification) from the scans that are performed. It is essential to understand the error properties of these techniques to know whether the calculated values are useful for the intended purpose.

To systematically evaluate these different turbulence measurement techniques, a Doppler lidar cycled each hour continuously through the methods during the last two weeks of the e**X**perimental **P**lanetary boundary layer **I**nstrument **A**ssessment
(XPIA) field campaign. These measurements are compared with sonic anemometer measurements at six heights on a 300 m meteorological tower located 540 m from the lidar. Through this comparison, the following questions will be addressed in this study.

– How accurate are the various single-Doppler turbulence measurement strategies in determining turbulence characteristics? Does the accuracy vary depending on the measurement height?

– What main caveats need to be considered when applying each technique? How should random errors and instrument noise be characterized and treated?

– What is the optimal operational scanning strategy to derive turbulence estimates? Should different strategies be used for different objectives?

To address these questions, the paper is organized as follows. An overview of the experiment and the instrumentation
used is detailed in Sect. 3. The various scanning strategies and methods to measure turbulence, including specific details of implementation, are described in Sect. 2. Within Sect. 4, the techniques are statistically compared through validation with sonic anemometry. Implications for future studies and possible future research directions are discussed within Sect. 5. A summary and conclusions are provided in Sect 6.

## 2 Turbulence measurement strategies

The scanning procedures used most often by Doppler lidars are azimuthal scanning, elevation scanning, and stares at one look angle. Each of these approaches can be used to measure one or more of the velocity variances and covariances. The theory for





turbulence measurements is based on the relationship between the observed radial velocity $v_r$ and the flow within the resolution volume given by

$$v_r = u\cos\theta\cos\phi + v\sin\theta\cos\phi + w\sin\phi + \epsilon, \tag{1}$$

wherein $u$ is the streamwise horizontal velocity, $v$ is the crosswise horizontal velocity, $w$ is the vertical velocity, $\phi$ is the elevation angle above the horizon, $\theta$ is the angle between $u$ and the azimuth of the lidar, and $\epsilon$ is uncorrelated random error in the measurement. The value of $\epsilon$ typically increases with range from the lidar, as the signal-to-noise ratio (SNR) decreases. By squaring Eq. 1 and removing the mean from each quantity, the radial velocity variance is given by

$$\overline{v_r'^2} = \overline{u'^2}\cos^2\theta\cos^2\phi + \overline{v'^2}\sin^2\theta\cos^2\phi + \overline{w'^2}\sin^2\phi + 2\overline{u'v'}\sin\theta\cos\theta\cos^2\phi + 2\overline{u'w'}\cos\theta\cos\phi\sin\phi + 2\overline{v'w'}\sin\theta\cos\phi\sin\phi + \overline{\epsilon^2}, \tag{2}$$

where the covariance terms involving $\epsilon$ are zero since it is uncorrelated. All of the turbulence measurements techniques are ultimately based on Eq. 2. Brief derivations and details of how these measurements are made, in addition to modifications introduced within this study, are described here. Complete derivations for each method can be found in the works cited.

## 2.1 Velocity azimuth display

While PPI scans have been used to take accurate measurements of the mean wind through VAD analysis (Smith et al., 2006), these scans can also be used to quantify turbulence. Eberhard et al. (1989) details a technique for measuring turbulence from PPI scans, based on pioneering work by Wilson (1970) and Kropfli (1986) wherein turbulence is measured using Doppler radar observations. From PPI scans at two sufficiently different elevation angles, all six components of the Reynolds stress tensor can be retrieved using the residuals of the VAD fitting by utilizing a partial Fourier decomposition of Eq. 2. However, the covariances and momentum fluxes $\overline{u'v'}$, $\overline{u'w'}$, and $\overline{v'w'}$ can be measured from any single PPI scan, and TKE can be obtained from a single scan if $\phi = 35.3°$ (for mathematical basis, see Eq. 4a in Eberhard et al., 1989).

A sample PPI scan for a turbulent time period is shown in Fig. 1a. For each range ring, the mean wind speed and direction are determined using VAD analysis. The complete VAD analysis described by Browning and Wexler (1968) includes terms for the vertical velocity of the scatterers as well as horizontal divergence, stretching deformation, and shearing deformation. However, a more simplified variation of the VAD analysis is often used by neglecting divergence and the deformation terms. For the results presented here, the simplified form is used since it yields more accurate estimates of the measured turbulent quantities when compared with sonic anemometer measurements. This may be due to variability from large turbulent motions being incorrectly partitioned into the divergence or deformation terms. However, in complex terrain or other locations these terms may not be negligible. An example of the fitting of this equation and its residuals $v_r'$, which are deviations from the expected mean $v_r$, is shown in Fig. 1b. If the mean flow (i.e., $\overline{u}$, $\overline{v}$, and $\overline{w}$) is homogeneous over the scanning circle, then the residuals of the fitting are results of turbulent motions and $\epsilon$. This is visualized within Fig. 1c, wherein coherent areas of positive and





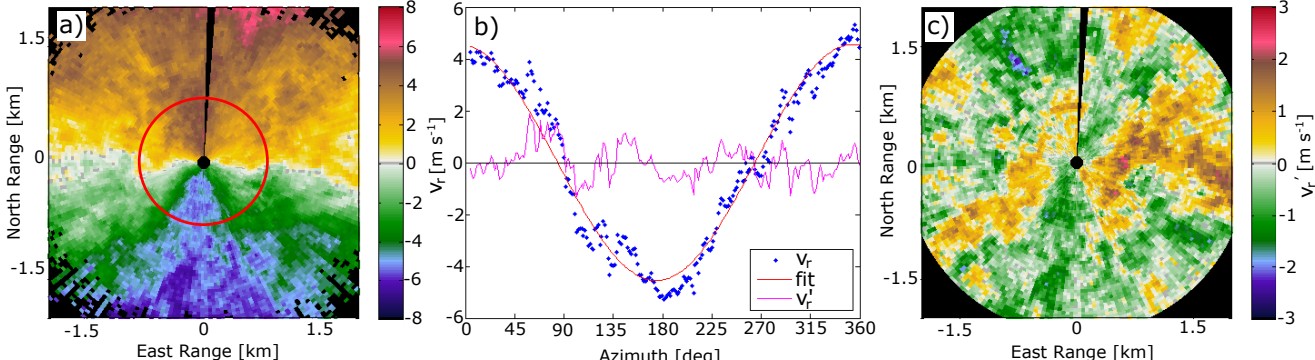

**Figure 1.** Sample PPI scan (a) during a turbulent time period, with the VAD fitting and its residuals to $v_r$ observations at the range ring denoted by the red circle shown in (b). Turbulence structures can be visualized in the residuals across the entire scan (c).

negative $v_r'$ represent turbulent eddies. Since turbulent structures are correlated spatially, $\epsilon$ can be quantified and removed by applying a structure function fit to the autocovariance of $v_r'$ across radials for a given range gate using the technique outlined by Lenschow et al. (2000). To our knowledge, this is the first time that the autocorrelation technique has been used to remove noise variance from a scan, as it is typically used for a timeseries from prolonged stares (Mayor et al., 1997). This technique

can lead to an overestimate of $\epsilon$ when the inertial subrange is smaller than the distance between adjacent azimuths, which is more likely at long ranges from the lidar as the spatial separation between adjacent beams increases.

Previously, measurements using the technique described by Eberhard et al. (1989) have not been evaluated against *in situ* observations. Wang et al. (2015) used a variation of this technique by applying it to a $30°$ sector PPI and assumed isotropic turbulence to relate $\overline{v_r'^2}$ to TKE. Estimates of TKE from the arc scan showed good agreement ($r^2 = 0.89$) with those from sonic

anemometer on a linear scale. Other studies (e.g., Berg et al., 2013; Sathe et al., 2015b) have used a loose variation of this VAD technique to quantify turbulence by using only a small number of beams ($4-6$) spaced around the entire $360°$, which is substantially different from using more than 100 beams around the sampling ring.

## 2.2   Six-beam

Sathe et al. (2015b) propose a technique to measure all six components of the Reynolds stress tensor by continuously cycling

between measurements at six different angles. One beam is vertical, and the other five are at a set elevation angle ($45°$ herein) and are equally spaced $72°$ apart in azimuth. For each beam, the time series of $v_r$ are linearly detrended over a fixed time window, which is 20-min here, and $v_r'$ is computed as its residual. A shorter window or a higher-order detrending may result in effectively removing large-scale eddies. Values of $\overline{v_r'^2}$ are computed for each beam separately. Thus, there are six known values





of $\overline{v_r'^2}$, one for each beam, and each is a function of differently weighted velocity variances and covariances based on the scan elevation and azimuth, as in Eq. 2. This can be represented by the matrix relationship

$$
M \begin{bmatrix} \overline{u'^2} \\ \overline{v'^2} \\ \overline{w'^2} \\ \overline{u'v'} \\ \overline{u'w'} \\ \overline{v'w'} \end{bmatrix} = \begin{bmatrix} \overline{v_{r1}'^2} \\ \overline{v_{r2}'^2} \\ \overline{v_{r3}'^2} \\ \overline{v_{r4}'^2} \\ \overline{v_{r5}'^2} \\ \overline{v_{r6}'^2} \end{bmatrix},
\tag{3}
$$

where $M$ is a 6 by 6 matrix of coefficients based on different combinations of $\theta$ and $\phi$, as in Eq. 2. Thus, it is possible to solve
for the six unknown components of the Reynolds stress tensor through an inversion of Eq. 3.

For each beam, the lidar stared at the given location for 1 s before advancing to the next position. Since data were collected at 2 Hz, two samples were collected 0.5 s apart. To remove uncorrelated noise $\epsilon^2$ from the observed $\overline{v_r'^2}$ for each beam, the autocovariance at the first lag for the samples that were 0.5 s apart was taken as $\overline{v_r'^2}$, following the technique presented by Lenschow et al. (2000). This likely results in a slight underestimate of $\overline{v_r'^2}$, since contributions from small eddies that are
uncorrelated over short timescales are removed. In the future, it is recommended that more samples be collected along each beam so that a structure-function or linear fitting may be applied to the autocovariance for a more robust measurement of $\overline{v_r'^2}$ for each beam. On average, the scanner took $\approx 3.6$ s to slew between beam positions, so that the scanner returned to the same beam every $\approx 27$ s.

### 2.3  RHI scans and vertical stares

Shallow RHI scans have also been used as a means to measure horizontal velocity variances (e.g., Banta et al., 2006; Pichugina et al., 2008b).These scans are conducted by scanning from the horizon up to $\approx 30°$, typically at two angles orthogonal to each other. Since the scans are mostly at low angles, it is assumed that the observed $v_r$ are due to the horizontal wind and that the contribution of $w$ is negligible. To ensure that measurements at different elevation angles are comparable, values of $v_r$ are normalized by $\phi$ by

$$
v_{rH} = \frac{v_r}{\cos\phi},
\tag{4}
$$

where $v_{rH}$ is the radial velocity projected in the horizontal. For each RHI, observations are binned by height (30-m bins used herein) which are used to make a mean profile of $v_{rH}$. The profile of $\overline{v_{rH}}$ is used to calculate deviations from the mean flow $v_{rH}'$. This is done by simply taking the difference between $v_{rH}$ and $\overline{v_{rH}}$ for the given height, where $\overline{v_{rH}}$ is linearly interpolated between the center of mass of each height grid. The variance of $v_{rH}'$ is calculated using the same height grid to produce a profile
of $\overline{v_{rH}'^2}$, which is the horizontal wind variance within the RHI plane. An example of this process and each derived product is provided in Fig. 2.




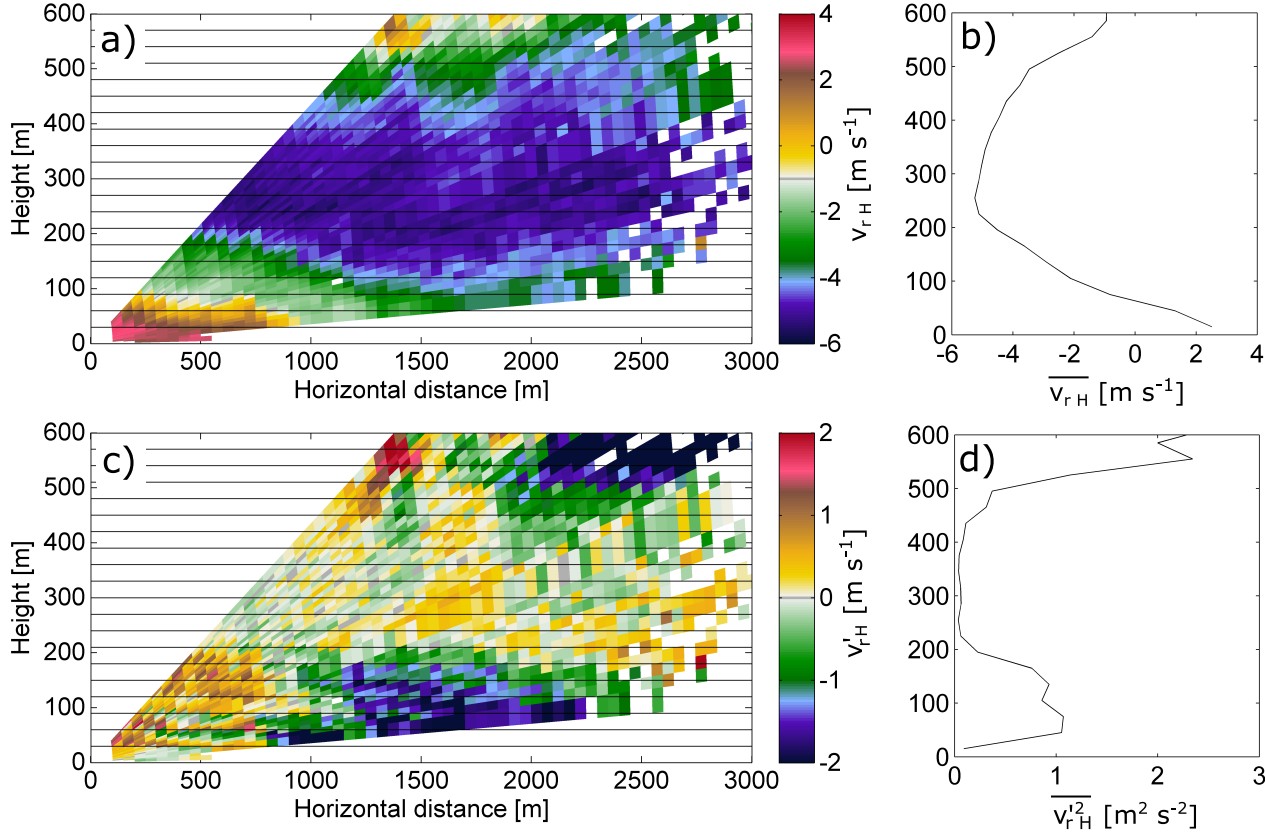

**Figure 2.** Sample RHI scan (a) showing instantaneous values of $v_{rH}$ over the scan plane. (b) Vertical profile of the mean $v_{rH}$ for the scan, which is used to calculate $v'_{rH}$ shown (c). For each height bin between the solid horizontal black lines on (a) and (c), the variance of $v'_{rH}$ is calculated resulting in the profile shown in (d).

When one of these scans is oriented with the mean flow and the other transversely, the two measured profiles of $\overline{v'^2_{rH}}$ can be treated as $\overline{u'^2}$ and $\overline{v'^2}$ respectively. If the scans are not oriented in such a way or if large directional shear is present, it is possible to rotate the variances to be aligned with the mean flow by

$$\overline{u'^2} = \overline{v'^2_{rH1}}\cos^2\Theta_1 + \overline{v'^2_{rH2}}\cos^2\Theta_2 - \overline{v'_{rH1}v'_{rH2}}\sin 2\Theta_1 \text{ and} \tag{5}$$

$$\overline{v'^2} = \overline{v'^2_{rH1}}\sin^2\Theta_1 + \overline{v'^2_{rH2}}\sin^2\Theta_2 + \overline{v'_{rH1}v'_{rH2}}\sin 2\Theta_1, \tag{6}$$

wherein $\Theta$ is the angle between the RHI scan azimuth and the mean flow, the subscripts denote the two different RHI scan planes, and the scans are orthogonal. Although the covariance term $\overline{v'_{rH1}v'_{rH2}}$ cannot be measured with this method, it becomes





**Table 1.** Summary of measured variables for each type of scanning strategy.

| | $\overline{u'^2}$ | $\overline{v'^2}$ | $\overline{w'^2}$ | $\overline{u'v'}$ | $\overline{u'w'}$ | $\overline{v'w'}$ | TKE |
|---|---|---|---|---|---|---|---|
| VAD (single $\phi$) | | | | ✓ | ✓ | ✓ | ✓ ($\phi = 35.3°$) |
| VAD (two $\phi$s) | ✓ | ✓ | ✓ | ✓ | ✓ | ✓ | ✓ |
| Six-beam | ✓ | ✓ | ✓ | ✓ | ✓ | ✓ | ✓ |
| RHI & vertical stare | ✓ | ✓ | ✓ | | | | ✓ |

zero if turbulence is assumed to be isotropic. Thus, values of $\overline{u'^2}$ and $\overline{v'^2}$ can be computed directly through the rotation. The mean wind profile, including wind speed and direction necessary for the rotation, are directly computed using the two profiles of $\overline{v_{rH}}$. Using this technique, there is no straightforward way to remove contamination from $\overline{\epsilon^2}$ in the variances. Thus, data were removed if the SNR$< -27$ dB to reduce contamination from highly noisy data.

To calculate TKE, values of $\overline{w'^2}$ also need to be known. For quantification of $\overline{w'^2}$, vertical stares were used in conjunction with the shallow RHI scans. Vertical stares are the most straightforward method to measure any vertical turbulent quantity with a Doppler lidar. Since the $w$ profile is continuously measured, it is simple to take the variance of time series of $w$ to obtain $\overline{w'^2}$. However, $\epsilon^2$ contaminates the measurement and needs to be removed to improve the accuracy of the measurement. As described earlier, the autocovariance technique described by Lenschow et al. (2000) is used to remove instrument noise.

Herein, values of $\sigma_w^2$ are taken as the extrapolated $-2/3$ structure function fit to the autocovariance of the time series at lags 1-5. Using this technique removes contamination by $\epsilon^2$ and mitigates volume averaging effects, which otherwise reduce the observed $\overline{w'^2}$ (Bonin et al., 2016).

## 3   Experimental overview

The e**X**perimental **P**lanetary boundary layer **I**nstrument **A**ssessment (XPIA) field campaign was conducted at the now-defunct

Boulder Atmospheric Observatory (BAO) in the spring of 2015. The primary goal of the XPIA experiment was to assess the capabilities for measuring flow in the planetary boundary layer with current technology such as wind profiling radars, microwave radiometers, sonic anemometers, various Doppler lidar systems, and other additional instrumentation. Herein, observations from a single scanning Doppler lidar system are compared with those from sonic anemometers on a meteorological tower. Pertinent details about the sensors and site are described here, but a complete list of the instrumentation and a thorough description

of the experiment along with its various objectives can be found in Lundquist et al. (2016).

The BAO was located in Erie, CO, approximately 25 km east of the foothills of the Rocky Mountains, and was designed primarily for PBL research as well as testing and calibration of various atmospheric sensors (Kaimal and Gaynor, 1983). Within the immediate vicinity, the terrain is relatively flat with gently rolling terrain. A 300-m tower was located on the property. For this experiment, 3-D Campbell CSAT3 sonic anemometers were installed on northwest (NW, 334°) and southeast (SE, 154°)

booms at six levels (50, 100, 150, 200, 250, and 300 m). Data were recorded at 20 Hz. A tilt-correcting algorithm that used a planar fit (Wilczak et al., 2001) was applied to the measurements after the experiment was finished. Data were filtered to




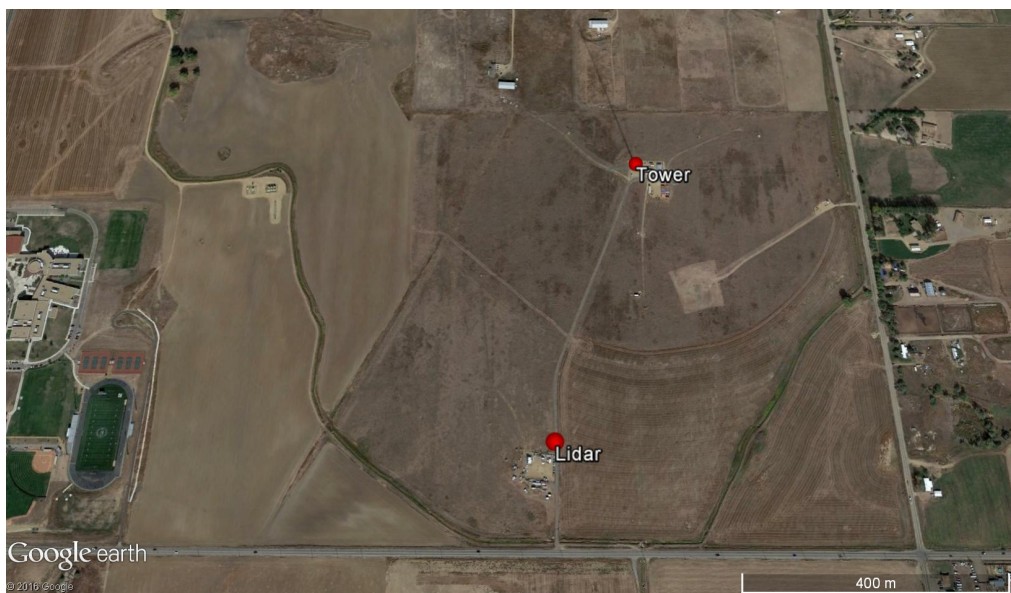

**Figure 3.** Satellite imagery of the BAO site with the locations of the 300-m tower and Doppler lidar deployment indicated.

remove time periods when the turbulence may be affected by the wake of the tower, following the results of McCaffrey et al. (2016). Specifically, data from the NW and SE sonics are removed when the wind direction is from between 100–170° and 300–20° respectively. Turbulence statistics from the sonic anemometers were averaged over 20-min blocks for comparison, similar to the averaging time for the various lidar scanning strategies discussed below. Any 20-min averages where the statistics

between the two sonic anemometers at the same height differed by a factor of 2 or more were removed, to ensure the statistics were comparable and not affected by the tower.

The modest complexity of the terrain and the proximity of the site to the mountains present complications in calculating turbulence quantities from remote-sensing data using the techniques described here. Under these conditions, the flow can exhibit non-turbulent variability along a scan that contribute to unknown degree to the calculated variances and covariances

producing larger variance and discrepancies between lidar and tower measured turbulent quantities. This variability is not expected over more homogeneous topography (e.g., Pichugina et al., 2008a) or the ocean (e.g. Tucker et al., 2009).

Measurements from only one Leosphere WINDCUBE 200S® are used in this study, although six scanning Doppler lidars were deployed during XPIA. At the end of the experiment from 15–31 May 2015, the system was deployed 540 m to the south-southwest of the 300-m tower, near the Visitor Compound, as shown in Fig. 3. The lidar was operating with a 50-m

pulse width and range gate size. The accumulation time for each beam was 0.5 s and the pulse repetition frequency (PRF) was 20 kHz. Due to the high PRF, the maximum unambiguous range of the lidar was only 7500 m. Hence, range folded echoes from clouds or other strong targets were occasionally apparent in the signal. These erroneous echoes were removed using a discontinuity-based algorithm described by Bonin and Brewer (2016). Each hour, a sequence of scanning strategies were conducted. For 20-min, the lidar cycled between PPI scans at a $\phi$ of 35.3° and 50.8° scanning at 3° s$^{-1}$ in azimuth. Over this





min period, five scans were completed at each elevation angle. Following the PPI scans, three shallow RHIs were performed at perpendicular angles ($\theta$ of 330° and 60°), followed by a 10-min vertical stare. For the rest of the hour, the six-beam scanning strategy was repeated, wherein each beam was sampled for 1 s before advancing to the next beam position as described in Sect. 2.2. This 1-hr scanning sequence was repeated continuously for the 17 days at the end of XPIA.

## 4 Turbulence statistics comparison

For most measurements, turbulent quantities measured by the Doppler lidar are not at precisely the same height as the sonic anemometers. This difference in measurement height is dependent on the type of scan. For instance, the range gate center for vertical stares is identical to the height the sonic anemometers at 100–300 m (50-m was below the minimum range). However, the closest range gate from the six-beam scan to the 300-m sonic was at 282.8 m. Thus, lidar-measured turbulent quantities are interpolated to the sonic anemometer heights.

Depending on the application or field of use, different turbulent quantities may be desired. Within the wind energy industry, turbulence intensity $TI$ calculated as

$$TI = \frac{\sqrt{\overline{u'^2}}}{U},\tag{7}$$

where $U$ is the mean horizontal wind speed, is most often used as it is a measure of the variability of the inflow into the turbine and affects the design requirements (International Electrotechnical Commission, 2005). In boundary-layer meteorology and air quality, TKE calculated as

$$TKE = \frac{1}{2}(\overline{u'^2} + \overline{v'^2} + \overline{w'^2})\tag{8}$$

is often used as a measure of the turbulent mixing in the atmosphere (Stull, 1988; Arya, 1999). Additionally, the covariance terms $\overline{u'w'}$ and $\overline{v'w'}$ are the momentum flux and are necessary to test and validate models of atmospheric flow. Since these measures of turbulence are most commonly used, they are the focus of this section. A complete statistical comparison of each measured variable is provided in Appendix A.

### 4.1 Turbulence Kinetic Energy

For the six-beam, VAD with multiple $\phi$, and RHI/vertical stare techniques, TKE was directly computed as the sum of the measured velocity variances using Eq. 8. As discussed within Sect. 2.1, TKE can be directly computed from a 35.3° $\phi$ scan without measuring $\overline{u'^2}$, $\overline{v'^2}$, or $\overline{w'^2}$ directly. From here onward, measured quantities from two PPI scans at different $\phi$ are referred to as 'VAD' measurements, and those from one PPI scan at 35.3° are 'VAD 35.3°' measurements.

A sample 24-hr time series of TKE is provided in Fig. 4 to demonstrate the ability of the different methods to capture temporal changes. The diurnal pattern of TKE decreasing overnight between 03:00–12:00 UTC is visible in measurements




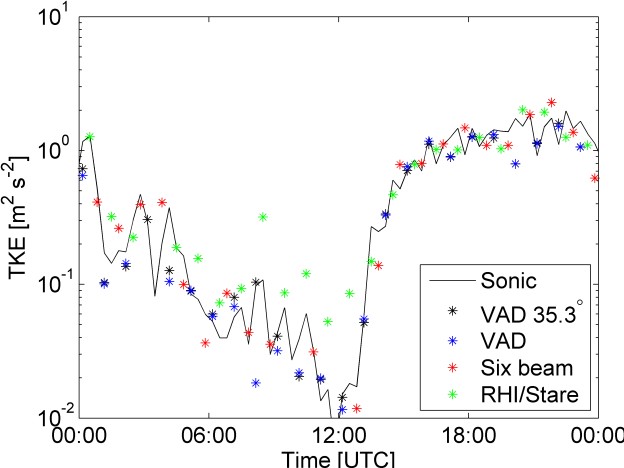

**Figure 4.** Sample time series of measured TKE on 30 May, 2015 at 200 m.

from the lidar and sonic anemometer. Although all the lidar techniques capture the decrease in TKE in the evening and early night hours (00:00–06:00 UTC), the TKE measurement from the RHI and vertical stare is systematically overestimated later in the night when TKE$< 0.1$ m$^2$ s$^{-2}$. The lidar measured TKE from the VAD, VAD 35.3°, and six-beam techniques capture the trends in TKE well. When TKE is small ($< 0.1$ m$^2$ s$^{-2}$) such as at 12:00 UTC, measured TKE by the six-beam technique can

be negative. Of all the TKE measurements during the experiment, 0.8% of the six-beam TKE values are negative. This result is unphysical, and is thought to be primarily due to sampling errors (see Lenschow et al., 1994) within each beam variance being propagated through the matrix inversion in Eq. 3, resulting in an underestimate of the variances. These sampling errors are difficult to quantify with the six-beam technique, since the time series is not continuous and the measurements are not equally spaced in time. Since negative TKE estimates are non-physical, they have been removed. The other methods analyzed did not

yield negative TKE values.

For a quantitative analysis of the TKE measurements, TKE values from the Doppler lidar and the southeast sonic anemometers are summarized in Fig. 5. These results are from measurements between 100–300 m for all the scanning strategies. No TKE measurements are available below 100 m since the first lidar range gate is 100 m so that no $\overline{w'^2}$ values are available below this height from the vertical stares. Comparisons with the northwest anemometers are similar and provide little additional in-

formation, thus are not shown. The comparisons shown here (and throughout the manuscript) are on a logarithmic scale since values can range several orders of magnitude, generally from 0.01–10 m$^2$ s$^{-2}$ for TKE. If the analysis were conducted on a linear scale, large values of TKE would dominate the best fit lines and small values would not be important. In logarithmic space, the ability to differentiate values of $\sim 0.01$ from $\sim 0.1$ is equally as important as differentiating $\sim 1$ from $\sim 10$, and





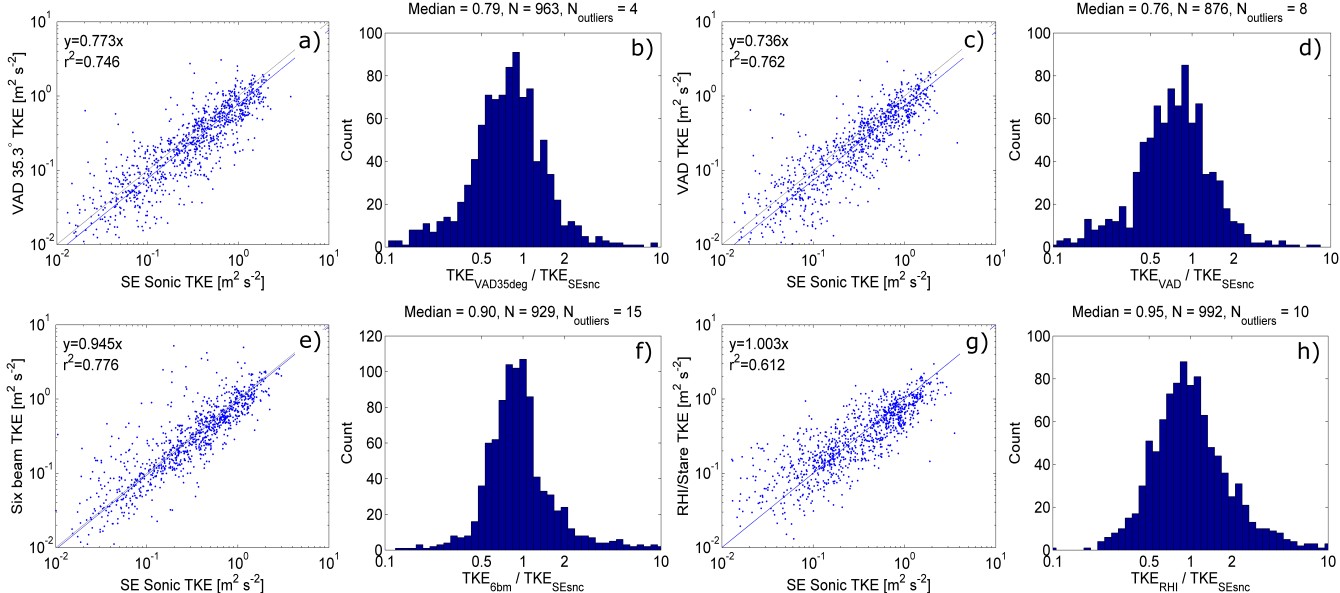

**Figure 5.** Scatter plots (a, c, e, g) and histograms (b, d, f, h) showing the relationship between the TKE measured by the lidar and southeast sonic anemometers at all heights. In a, c, e, g, the blue line is the best fit line given by the equation in the upper left and the black line indicates a 1-1 relationship. Histograms (b, d, f, h) show the ratio of the lidar measured TKE to the sonic measured TKE with the median ratio, number of points N, and number of outliers $N_{outliers}$ that are more than 1 order of magnitude apart. Lidar measurements are from TKE at $\phi = 35.3°$ (a, b), TKE from two $\phi$ (c, d), the six-beam technique (e, f), and the RHI and stare combination (g, h).

values across all different orders of magnitude are weighted equally in determining the trend line. Within each of the scatter plots in Fig. 5, the best fit lines were determined fitting

$$\log_{10}(y) = \log_{10}(x) + b, \tag{9}$$

where $x$ and $y$ are data points on their respective axes in linear units and $b$ is a constant. Transforming the equation back into

5    linear space for ease of interpretation, the equation shown in the upper left of Fig. 5a, c, e, g is $y = bx$.

Each of the techniques evaluated herein generally shows skill in measuring TKE, as indicated by $r^2$ values greater than 0.6 in Fig. 5. Considering that the sonic anemometer and lidar measurements represent different spatial areas, which vary according to the scanning technique, and that each are subject to sampling error, the authors consider the correlation between the lidar and sonic TKE to be good. Quantifying the sampling error would allow a more statistical determination of it measurements are

10    in agreement within their respective uncertainties. However, no techniques that the authors are aware of can be used to estimate the sampling error from any of the scan types except the vertical stares (see Lenschow et al., 1994). Due to these limitations and for consistency, sampling errors are not estimated for any of the scans. Instead, the relative correlations between sonic





and lidar measurements, and their differences for various scan strategies, are used to understand biases and accuracy of each technique.

The six-beam technique demonstrates the best ability to measure TKE overall, as evident by the largest $r^2$ and slope of 0.945, close to unity. Additionally, the histogram of the ratio of TKE measurements in Fig. 5f shows a distinct peak around 1 with

reduced spread compared to Fig. 5b, d, h; 83.5% of six-beam TKE values are within a factor of 2 of the sonic measurement, the largest proportion of all the techniques analyzed. However, the six-beam technique also produces the largest number of TKE outliers, defined as being more than 1 order of magnitude different from the sonic observed TKE. Approximately half of these outliers are negative TKE values, which were removed as discussed earlier. The other outliers are when TKE is grossly overestimated, as visible in Fig. 5e. Upon manual inspection of these high outliers, many are due to contamination of

range folded echoes. When range folded echoes appear intermittently, the $v_r$ time series within each beam position changes erratically, resulting in an anomalously large variance, and spuriously increasing the observed TKE. The discontinuity-based algorithm used to detect range folded echoes largely relies on contextual information from proximate beams in time and space (Bonin and Brewer, 2016) not available from the six-beam technique. These anomalous echoes can typically be detected and removed in PPI, RHI, and stare scans, but these range folded returns persist through the quality-control process of the six-beam

measurements and degrade the accuracy of the calculated variances .

As discussed in Sect. 2.1, the VAD technique can be used to measure TKE from either one scan at 35.3° or two scans at different $\phi$, herein 35.3° and 50.8°. Since the 35.3° scan is used in both approaches, the results from both methods are not independent of each other. This can be seen by the similar results from both approaches in Fig. 5a-d. Although the TKE from both techniques is highly correlated with sonic TKE, the VAD and VAD 35.3° measured TKE is systematically biased too low.

The low bias is more pronounced for the two-$\phi$ technique, although the scatter is reduced slightly as evidenced by the larger $r^2$ in Fig. 5c. This reduced scatter is attributed to smaller sampling errors, since twice the amount of data go into the measurement. The overall low biases may be due to the inability to capture all the scales of turbulence; the largest eddies may not be fully captured and resolved within the scanning circle. This effect would be more pronounced for higher elevation PPIs, such as at 50.8°, and explain the more significant low bias in TKE for the two PPIs used herein. If a lower $\phi$ were used (i.e., at 25°),

the low bias may not be as pronounced for the two $\phi$ VAD TKE. Unfortunately, no data are available from this experiment to validate this hypothesis. Despite these biases, 73.3% (35.3° VAD) and 71.7% (two VADs) of the TKE measurements are within a factor of 2 of the sonic TKE.

Using the RHI and vertical stares to measure TKE results in the largest scatter. Still, 74.7% of the lidar measured TKE are within a factor of 2 of the sonic measurements, indicating that the technique is still accurate. The six-beam and VAD techniques

show similar scatter for all ranges of TKE, and the RHI/stare technique typically overestimates TKE when its value is small (i.e., $< 0.1$ m$^2$ s$^{-2}$), as apparent in Fig. 5g. The cause of the overestimate during weakly turbulent conditions is unclear, but it may be due to spatial variability of the flow that the sonic cannot detect or the inability to remove instrument noise effects. These random errors are quantified and removed in the VAD and six-beam techniques as detailed in Sect. 2, but no established technique exists to remove these errors from RHI measurements, which may be addressed in a future study. Within Fig. 5h,

this high bias under weakly turbulent conditions manifests itself as right-skewed distribution.





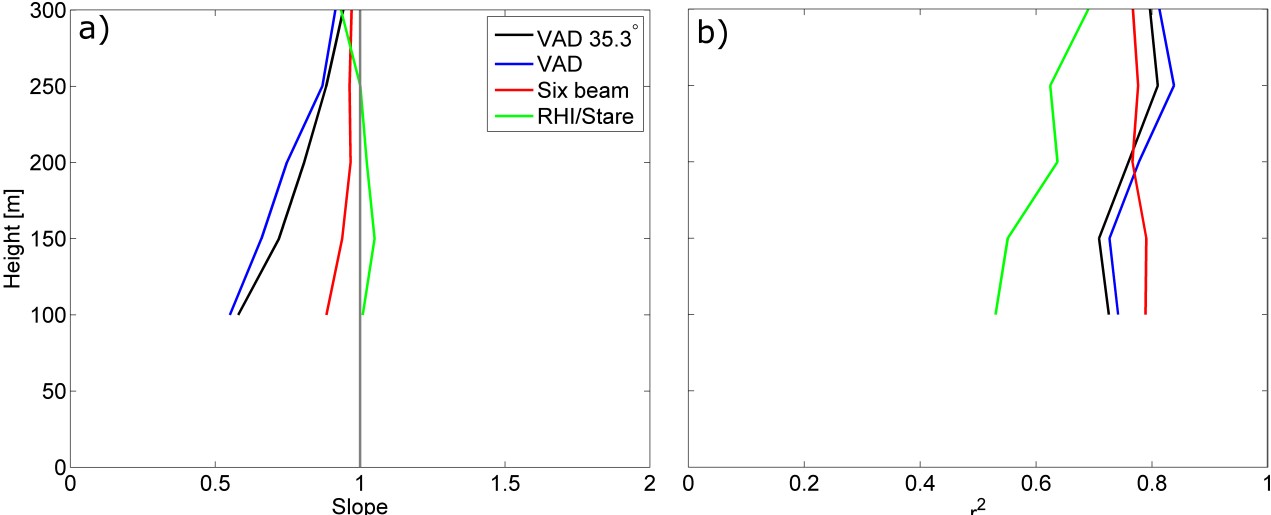

**Figure 6.** The slope (a) and $r^2$ (b) of the best fit line on a logarithmic scale (similar to those in Fig. 5) relating TKE measurements from the sonic anemometer and lidar for each measurement height.

The results shown in Fig. 5 are for all measurement heights combined. The analysis can be further refined by comparing lidar TKE measurements at each sonic anemometer height separately. Figure. 6 summarizes this analysis by showing the slope and $r^2$ of the best fit line at each height. The accuracy of the six-beam technique is the most consistent at all measurement heights, as the slope and $r^2$ are nearly constant with height. The value of $r^2$ remains around 0.75 at every height, whereas the slope increases a small amount with height. This change in slope indicates that TKE is less-underestimated above 200 m than it is closer to the surface.

The bias of TKE measurements using the RHI and vertical stare method is independent of height and small (Fig. 5a), as the slope is generally around 1. The TKE measurement becomes more accurate with height, indicated by the increase of $r^2$ with height in Fig. 5b. The cause for the increase in accuracy with height is unclear. The low bias of VAD and VAD 35.3° TKE observations becomes less significant with height, as shown in Fig. 5a. Coincidentally, the VAD and VAD 35.3° $r^2$ values increase with height, representing less scatter and more accurate values at higher altitudes.

To examine the decrease in low bias and increase in accuracy of VAD TKE measurements with height, the VAD circle diameter and typical largest eddy size, the integral length scale $l$, are compared. First, the integral time scale $t_{int}$ is calculated from a linearly detrended 20-min timeseries of $u$ from the sonic anemometer as

$$t_{int} = \frac{1}{\overline{u'^2}} \int_0^{\tau(A=0)} A(\tau)d\tau, \tag{10}$$

where $A(\tau)$ is the autocovariance of $u$, which is a function of the time lag $\tau$. Since the time series is discrete, $d\tau$ is the sampling interval (0.05 s here, since the sonic data rate is 20 Hz). The median and various percentile values of $l$, computed as $l = Ut_{int}$,



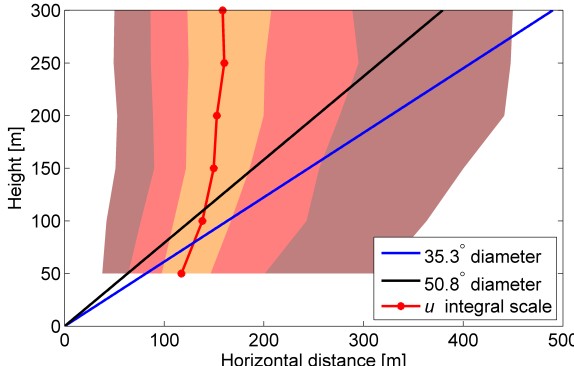

**Figure 7.** The diameter of the scanning circle of the two PPI scans is shown in comparison to $l$. The red line denotes the median $l$, while the progressively darker contours represent the 40–60%, 25–75%, and 10–90% percentile intervals of $l$ over the entire 17-day experimental period.

are shown as a function of height and with reference to PPI scan diameters in Fig. 7. Generally, individual 35.3° and 50.8° PPI scans do not fully sample the largest turbulent eddies close the ground, since the scan circle diameter is often less than $l$. The largest eddies are better captured by these scans at higher altitudes, especially for the 35.3° PPI scan. At 300-m, the integral scale is less than the 35.3° scan diameter over 90% of the time.

5    The results shown in Fig. 7 explain why VAD and VAD 35.3° TKE measurements become more accurate and less biased with height, as the largest turbulence scales are better captured. These effects are not important to the RHI method, since the spatial extent of the average is typically several km, much larger than the typical eddy size. Since the vertical stare and six-beam techniques use time series analysis, the largest scales of turbulence are observed if the time window length exceeds the integral time scale, which is often $\sim 10 - -100$ s during daytime (Lenschow et al., 2000; Bonin et al., 2016).

## 4.2    Turbulence Intensity

Similar to the analysis of TKE presented in Sect. 4.1, measurements of TI measured from the six-beam, VAD (using 2 $\phi$ angles), and RHI techniques are compared and validated here. Since TI $\to \infty$ as $U \to 0$ following Eq. 7, TI is only calculated when $U > 1$ m s$^{-1}$. A sample time series of TI is shown in Fig. 8. The diurnal trend in TI is clearly visible in the sonic measurements, as TI is generally low (3–10%) at night until 12:00 UTC and TI is larger (20–70%) during the day. During

15   the morning hours (i.e.,12:00–18:00 UTC), $U$ was less than 2.5 m s$^{-1}$, causing TI to become large. Despite some scatter, TI measurements from the Doppler lidar show a similar trend with smaller TI values at night and larger ones during the day.

Nonphysical negative $\overline{u'^2}$ values due to computational artifacts as described previously by Newman et al. (2016b) have been removed from the analysis. Measurements of TI at all heights over the entire experiment are summarized in Fig. 9. For each of the three techniques analyzed, the $r^2$ for TI is $\approx 0.2$ lower than it is for TKE. This indicates the combined velocity variance

20   components in TKE are more accurately measured than individual velocity variances separately (see also Table 2 for $\overline{u'^2}$, $\overline{v'^2}$,





and $\overline{w'^2}$ comparison statistics). Still, the VAD, six-beam, and RHI techniques each show skill in measuring TI. The VAD and six-beam techniques perform comparably, having a similar $r^2$ and slope indicated a low bias. Sathe et al. (2015b) also show that the six-beam technique tends to underestimate $\overline{u'^2}$. The RHI TI measurements show more scatter than the other two methods, given the lower $r^2$, but showed little bias.

5     The slope and $r^2$ of the best fit line as a function of height is shown in Fig. 10. Similarly to TKE, VAD measurements of TI are biased low near the ground as indicated by the slope of $\approx 0.7$ at 100 m. By 250 m, the bias becomes small as most of the turbulence scales are resolved, as discussed in Sect. 4.1. The accuracy of the VAD TI measurement does not change significantly with height, as $r^2$ does not consistently depend on height. The six-beam TI measurement is biased consistently low regardless of height, as indicated by the slope of $\approx 0.83$ at all heights, and the scatter of the measurements does not have

10  a consistent trend with height. The slope of RHI TI tends to be larger near the ground and slowly decrease with height, as evidenced in Fig. 10a. Coincidently, the scatter associated with these measurements decreases significantly with height, as the $r^2$ increases from 0.12 to 0.56.

## 4.3   Shear Velocity

The momentum flux terms $\overline{u'w'}$ and $\overline{v'w'}$ can be combined through the calculation of a stress velocity scale $u_*$ by

$$15 \quad u_* = \left[ \left( \overline{u'w'}^2 + \overline{v'w'}^2 \right) \right]^{1/4}. \tag{11}$$

Of the techniques analyzed, only the six-beam and VAD methods have a theoretical basis for measuring the covariances $\overline{u'w'}$ and $\overline{v'w'}$ necessary to compute $u_*$. Each PPI scan at any $\phi$ can independently provide a measurement of the covariances, so the $u_*$ values shown here are taken as the average of all PPI scan at both 35.3° and 50.8°. An example of a 24-hr time series

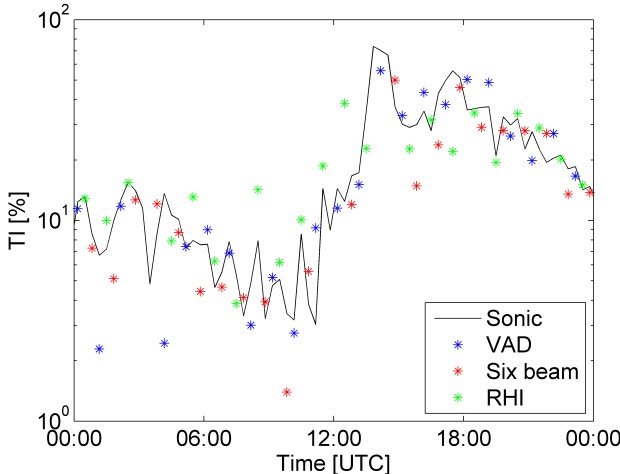

**Figure 8.** Sample time series of measured TI on 30 May, 2015 at 200 m.





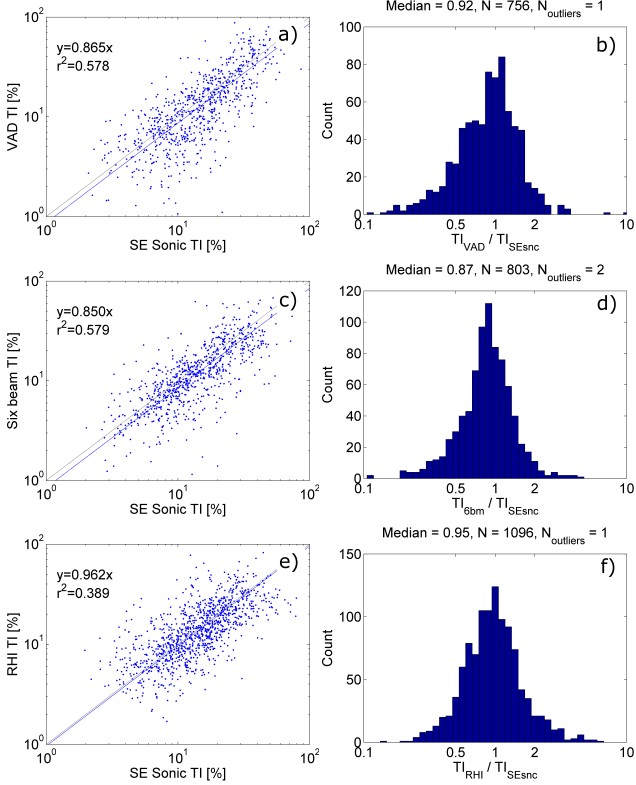

**Figure 9.** Same as Fig. 5 for TI instead of TKE. Lidar measurements are from TKE from two $\phi$ (a, b), the six-beam technique (c, d), and the RHI scans (e, f).

of $u_*$ is shown in Fig. 11. The sonic data are not shown for 00:00–03:00 UTC and 14:00 UTC, since the $u_*$ measurements on opposing booms were more than a factor of two different from each other, even though neither sonic was waked. Thus, neither is taken as a baseline measurement. For this sample period, the lidar and sonic data show a similar trend, values of $u_*$ decreasing for 00:00–12:00 UTC, rapidly increasing for 12:00–15:00 UTC, and remaining nearly constant after 15:00 UTC.

5 These trends are a result of $u_*$ steadily decreasing overnight, increasing in the morning hours, and remaining steady over the day.

The comparison between sonic and lidar $u_*$ measurements are summarized in Fig. 12. Both the VAD and six-beam techniques generally overestimate $u_*$, as shown in both the histograms and scatter plots. During time periods when the sonic estimated $u_*$ is small (i.e., $< 0.1$ m s$^{-1}$), the lidar techniques predominately overestimate $u_*$ as indicated by the large number

10 of data points above the one-to-one line in Fig. 12a, c. The small $r^2$ for the best fit lines indicates that there is substantial scatter in the comparison of the lidar and sonic measurements. Thus, the six-beam and VAD methods show little skill in being able to accurately measure $u_*$ and the covariances, as shown in Table 2. These results do not significantly change with height (not shown), as the $r^2$ remains small for both methods at all measurement heights between 50–300 m. The accuracy of covariance




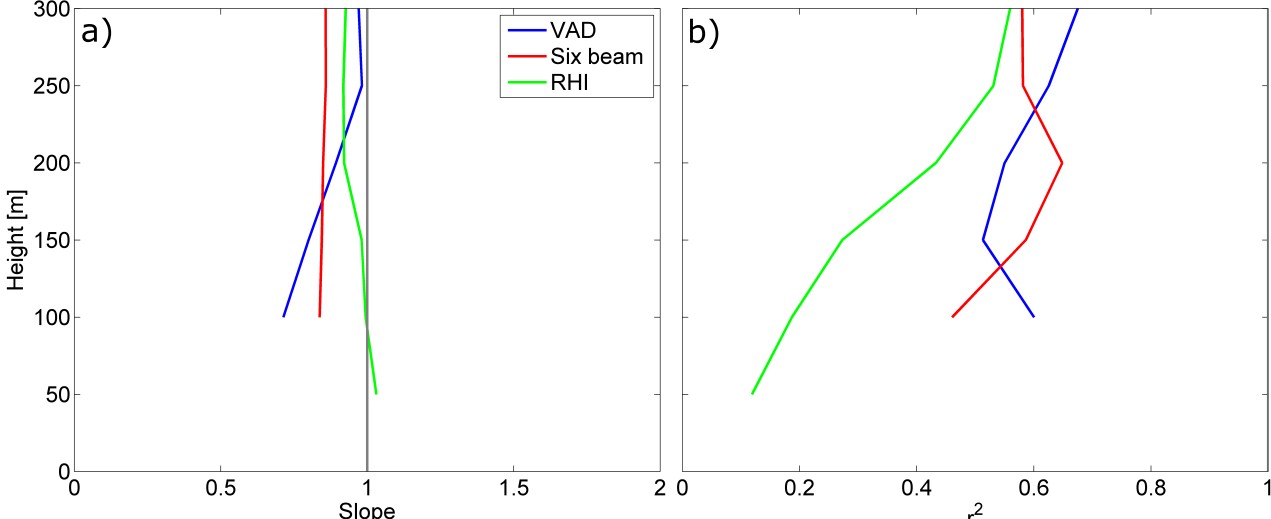

**Figure 10.** Same as Fig. 6 for TI instead of TKE.

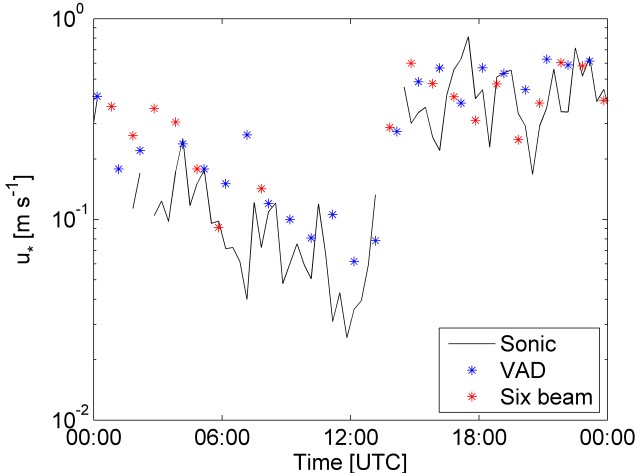

**Figure 11.** Sample time series of measured $u_*$ on 30 May, 2015 at 200 m.

and $u_*$ measurements from the VAD and six-beam methods has not been evaluated in the past, but here the measurements are found to exhibit large error. Over simpler topography, Berg et al. (2013) present results from a DBS technique that produced more accurate measurements of $\overline{u'w'}$ and $\overline{v'w'}$.





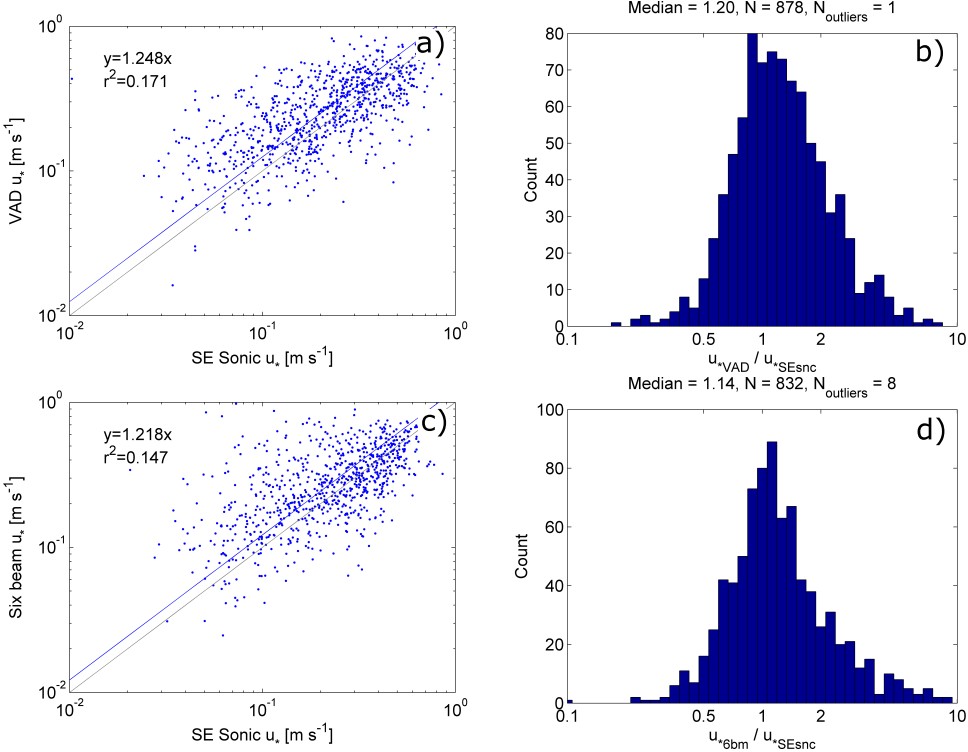

**Figure 12.** Same as Fig. 5 for $u_*$ instead of TKE. Lidar measurements are from TKE from two $\phi$ (a, b) and six-beam technique (c, d).

## 5 Discussion

From the results shown in Sect. 4, it is clear that TKE can be measured by each of the three techniques analyzed. However, measurements of each individual term of the Reynolds stress tensor are more difficult to accurately measure. The velocity covariances are particularly difficult to quantify, as the six-beam and VAD techniques show little skill in their measurement.

5   It is thought that the poor comparison for the covariance terms is due to the fact that the sampling error for the measurement exceeds the covariance typical dynamic range. Based on sonic anemometer observations, 80% of $|\overline{u'v'}|$, $|\overline{u'w'}|$, and $|\overline{v'w'}|$ were $< 0.1 \, \mathrm{m^2 \, s^{-2}}$. Also, covariance terms having small correlations take much longer to converge to a stable value (Lenschow et al., 1994).

### 5.1 Strengths and limitations of each strategy

10   Each of the scanning strategies evaluated herein has its own strengths and limitations. One of the biggest limitations for all of the techniques except vertical stares is that turbulence is assumed to be homogeneous over the area of each scan. Thus, these techniques do not always work well in complex terrain or differential land use where turbulence can significantly vary spatially. For the VAD and RHI techniques in particular, spatial variations in the mean wind due to local drainage flows (e.g.,





Banta et al., 1997; Choukulkar et al., 2012) can result in large deviations from the spatially averaged mean wind. Since these methods are unable to differentiate turbulent deviations from mean deviations, turbulence is overestimated. In these situations, it may be possible to use arc segments from the PPI scans to compute TKE (Wang et al., 2015) over different radials where the mean flow is homogeneous. With the current technology, multi-Doppler measurements (i.e., Mann et al., 2009; Fuertes et al., 2014; Newman et al., 2016a) are best able to quantify turbulence at specific locations in complex terrain.

The spatial height resolution for the PPI, six-beam, vertical stare, and RHI scans largely depends on scan geometry. Direct measurements of $\overline{w'^2}$ from vertical stares can only be taken starting at the height of the lowest range gate and the spatial resolution is limited to the range gate size. Since the six-beam technique presented here and in previous studies (e.g., Sathe et al., 2015b; Newman et al., 2016b) uses a vertical beam, the spatial resolution is limited by that beam the same as vertical stares. Future studies may try removing the vertical beam and instead use six-beams all at $\phi = 45°$, or another $\phi$, to take measurements at a lower altitude. The vertical resolution of a PPI scan is dictated by its $\phi$: a larger $\phi$ results in a higher minimum measurement height, reduced vertical resolution, greater height coverage, and reduced horizontal scan footprint compared to typical eddy size. The residuals in the PPI scans are more sensitive to $\overline{w'^2}$ for a larger $\phi$, and are more sensitive to $\overline{u'^2}$ and $\overline{v'^2}$ for a smaller $\phi$. The height resolution of an RHI scan is truly customizable, as $\overline{u'^2}$ and $\overline{v'^2}$ are computed by user-defined height bins. Since RHI scans typically start or end at the horizon, $\overline{u'^2}$ and $\overline{v'^2}$ can be calculated within a few meters of the surface. On the other hand, this technique is especially susceptible to non-turbulent horizontal variability along the scan due to complex terrain and other effects, especially since small $\phi$ that cover large distances horizontally are used.

Although a 20-min averaging interval is used here for comparison, measurements for several of the techniques could be made much quicker. Turbulence statistics from vertical stares and the six-beam technique are computed through typical time series analysis; thus, the time series needs to be long enough to ensure that the largest turbulent eddies pass through the resolution volume to be captured (see Lenschow et al., 1994), yet short enough that the flow can be considered stationary (Banta et al., 2013). Since the VAD and RHI methods compute turbulence through quantifying spatial variability, they are not subject to these same sampling error limitations when all scales of turbulence are captured in the scanning volume. With a data rate of 2 Hz, each PPI scan with 240 beams takes 2 min to complete; thus TKE can be measured in 2 min with a scan at $\phi = 35.3°$ and each velocity variance and covariance can be measured from two scans, taking 4 min. Since each RHI scan can be conducted in $< 1$ min, $\overline{u'^2}$ and $\overline{v'^2}$ can be measured in $\approx 2$ min.

The methods presented here measure velocity variances, but none currently are able to distinguish atmospheric turbulence from submeso motions, including waves. Since the value of TKE is calculated from $\overline{u'^2}$, $\overline{v'^2}$, and $\overline{w'^2}$, the observed TKE may be a mixture of turbulent and submeso variances and not always a measure of pure atmospheric turbulence. Considering these submeso motions have been predominantly documented within the nocturnal stable PBL when turbulence is typically weak (Mahrt, 2014), the value of TKE defined as a measure of turbulent motion may be overestimated when waves are present. Numerically differentiating between non-turbulent and turbulent motions is difficult (Stewart, 1969) and is best done through multiresolution decomposition or wavelet analysis (e.g., Cuxart et al., 2002; Vickers and Mahrt, 2003; Viana et al., 2010). This requires a high-resolution ($> 1$ Hz) time series. Thus, out of the methods analyzed, only the vertical stare has the data necessary separate turbulent from non-turbulent motions using established techniques.





## 5.2 Future directions for improving turbulence estimates

One of the main limitations of the six-beam technique using current commercially available scanning Doppler lidars is the return time between samples at the same beam position ($\approx 27$ s). When the six-beam technique was performed, the scanner spent 78% of the time slewing from one beam to the next. Thus a 2-axis hemispheric scanner is not be the best option for

running the six-beam technique. A wedge scanner that can quickly rotate between beam positions is more appropriate, as the time between beams could be minimized. This would yield a higher temporal resolution time series for each beam, enabling a better method of noise removal through a structure function fit (Lenschow et al., 2000) and possibly differentiating between turbulent and non-turbulent variances through multiresolution decomposition or wavelet analysis.

The RHI technique is best suited for measuring $\overline{u'^2}$ and $\overline{v'^2}$ near the surface ($< 100$ m), since the measurements need to be

made at a low angle. However, there is currently no method to remove random errors from RHI measurements. Thus, it may be better in the future to simply perform shallow horizontal stares where $\phi < 20°$ to measure the horizontal variances. Removing noise and correcting for volume averaging effects would be straightforward (Bonin et al., 2016), and it also may be able to distinguish turbulence from non-turbulent motions.

## 6    Conclusions

The XPIA field experiment was conducted at the Boulder Atmospheric Observatory in the spring of 2015. For 17 days at the end of the experiment, a Leosphere WINDCUBE 200S® continuously alternated between a PPI, RHI, vertical stare, and six-beam scanning strategy. Measurements from each scan type were used to calculate components of the Reynolds stress tensor and other measures of turbulence. These Doppler lidar turbulence measurements were compared to those from sonic anemometers on a 300-m tower located 540 m from the lidar to evaluate the accuracy of each technique.

Overall, TKE and velocity variances (i.e., $\overline{u'^2}$, $\overline{v'^2}$, $\overline{w'^2}$) were more accurately measured by the six-beam technique. Six-beam measurements showed the best agreement with the sonic-anemometer data across all ranges of turbulence magnitude. Additionally, the error and bias of the six-beam turbulence measurements did not significantly change with height. On the other hand, the VAD measurements of TKE and velocity variances tended to become more accurate with height. VAD-measured turbulence tended to be biased low near the surface, and this bias decreased with height. This bias is attributed to the inability

of the PPI scan to resolve all scales of turbulence near the surface, since the largest eddies extend beyond the scanning circle. The scanning volume geometrically becomes larger with height; thus, the PPI is better able to resolve all scales of turbulence and make more accurate measurements of turbulent quantities farther from the surface. Although the sonic anemometer observations agreed most poorly with RHI-measured TKE and TI, it showed little bias (slope of linear regression for TKE was 1.003) and still showed considerable skill in measuring turbulence. The inability to quantify and remove random errors from

the RHI measurements led to an overestimate under time periods when turbulence was weak (TKE$< 0.1$ m$^2$ s$^{-2}$). The methods evaluated herein showed little skill in measuring $u_*$ and velocity covariances.

When selecting a scanning strategy in future experiment, one needs to consider the desired turbulence measurements. While the RHI technique may be the least accurate of the three evaluated, it is the only method that can obtain measurements just



above the surface. If a rapid update time is desired (i.e., $< 5$ min), the VAD technique may best address these needs. Vertical stares and the six-beam technique use time series analysis to quantify turbulence. If the temporal resolution is sufficiently high, established techniques may be used to partition turbulent and non-turbulent variance, which no method currently exists for the RHI and VAD data.

**Appendix A: Comparison of all measured turbulent quantities**

A complete statistical comparison of all measures of turbulence is provided for reference in Table 2. For brevity, only a selected portion of these results were closely evaluated within the main body of the manuscript. While the results in Table 2 summarize measurements at all heights, the accuracy and bias of velocity variances and covariances as a function of height are similar to those presented in Sect. 4. The results have been summarized through comparisons of values on both a logarithmic scale

(except for the covariances which can be negative) and a linear scale.

*Acknowledgements.* The authors acknowledge financial support for this work was provided by the U.S. Department of Energy, Office of Energy Efficiency and Renewable Energy, the NIST Greenhouse Gas Measurements and Climate Research Program, and NOAA's Earth System Research Laboratory. The authors are also thankful to numerous individuals and organizations who assisted with the field deployment and processing of the data including Tom Ayers, Bruce Bartram, Duane Hazen, Paul Johnston, Jesse Leach, Katherine McCaffrey, Lefthand

Water District, Erie High School, and the St. Vrain School District. We also appreciate NOAA/Earth Systems Research Laboratory/Physical Sciences Division for supporting the instrumentation at the BAO. We express appreciation to the National Science Foundation for supporting the CALB deployments of the tower instrumentation (https://www.eol.ucar.edu/field_projects/cabl).





**Table 2.** Statistical comparison between sonic anemometer and lidar observations of all measured turbulence variables. Slope and $r^2$ values are computed seperately for regression analysis on a logarithmic and linear scale. The column '% Non-physical' indicates the % of measurements that are negative or non-real.

| Variable | Method | Slope (log) | $r^2$ (log) | Slope (linear) | $r^2$ (linear) | % Non-physical |
|---|---|---|---|---|---|---|
| TKE | VAD ($2\,\phi$) | 0.736 | 0.762 | 0.721 | 0.614 | 0 |
| | VAD ($\phi = 35.3°$) | 0.773 | 0.746 | 0.731 | 0.559 | 0 |
| | Six-beam | 0.945 | 0.776 | 0.913 | 0.562 | 0.8 |
| | RHI/Vertical Stare | 1.003 | 0.612 | 0.791 | 0.632 | 0 |
| $\overline{u'^2}$ | VAD ($2\,\phi$) | 0.940 | 0.524 | 0.876 | 0.335 | 7.9 |
| | Six-beam | 0.872 | 0.620 | 0.835 | 0.342 | 8.1 |
| | RHI | 0.918 | 0.355 | 0.627 | 0.294 | 0 |
| $\overline{v'^2}$ | VAD ($2\,\phi$) | 0.977 | 0.620 | 0.910 | 0.393 | 4.9 |
| | Six-beam | 0.914 | 0.680 | 1.033 | 0.568 | 8.8 |
| | RHI | 0.981 | 0.372 | 0.679 | 0.314 | 0 |
| $\overline{w'^2}$ | VAD ($2\,\phi$) | 0.739 | 0.542 | 0.539 | 0.268 | 35.4 |
| | Six-beam | 0.996 | 0.789 | 0.927 | 0.648 | 0 |
| | Vertical Stare | 0.971 | 0.790 | 0.979 | 0.591 | 0 |
| $\overline{u'v'}$ | VAD | | | 0.150 | 0.005 | 0 |
| | Six-beam | | | 0.006 | 0.001 | 0 |
| $\overline{u'w'}$ | VAD | | | -0.165 | 0.007 | 0 |
| | Six-beam | | | -0.020 | 0.005 | 0 |
| $\overline{v'w'}$ | VAD | | | -0.010 | 0.001 | 0 |
| | Six-beam | | | 0.001 | 0.002 | 0 |
| TI | VAD ($2\,\phi$) | 0.865 | 0.578 | 0.967 | 0.535 | 7.9 |
| | Six-beam | 0.850 | 0.579nb | 0.837 | 0.505 | 8.1 |
| | RHI | 0.962 | 0.389 | 0.865 | 0.248 | 0 |
| $u_*$ | VAD | 1.248 | 0.171 | 0.992 | 0.133 | 0 |
| | Six-beam | 1.218 | 0.147 | 1.061 | 0.089 | 0 |

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
