# Peer review of "Evaluation of Turbulence Measurement Techniques from a Single Doppler Lidar"

_Atmospheric Measurement Techniques, 2017_

## Referee Comment (RC1) · Anonymous Referee #1 · 8 Mar 2017

General comments The paper presents observations from the eXperimental Planetary boundary layer Instrumentation Assessment (XPIA). The observations were used to verify Doppler lidar turbulence profiles through comparison with sonic anemometer measurements. During a 17-days period, a single scanning Doppler lidar continuously cycled through different turbulence measurement strategies: velocity azimuth display, six-beam, and range height indicators with a vertical stare. The investigation focused on turbulence kinetic energy, turbulence intensity, and shear velocity data. For evaluation, sonic anemometer measurements at six heights on a 300-m tower were available. The paper is well written and of general interest for the lidar community (scientists and users alike). I therefore recommend the paper to be published with minor revisions.

specific comments Section 1: the introduction is quite general and could be more concise on the topic tackled (more focused) in this investigation. Page 5, line 2: Specify

which of the methods mentioned in Lenschow et al. (2000) is used to remove noise. Page 5, line 17: could you make a statement concerning the time window for detrending (15, 20, 30, 60 min?) on the turbulence results. Applying a 20-min window could filter out large convective cells. Page 9, line 14: '... 50-m pulse width ... ': does that mean that physically independent measurements are (physical resolution of the lidar) 50 m? Page 12, line 5: y = bx: Transformation of equation (9) gives y = x 10b Figure 5 and 9: a zero line would be helpful. Page 13, line 22 (" ....... may be due to the inability to capture all the scales of turbulence"): Spectra should be included in order to see which scales are not captured (to prove the statement would be good) Page 15, line 18: (negative u-variance values). Is this the same effect as for TKE mentioned on page 11, line 5? Page 16, line 7: "the bias becomes small as most of the turbulence scales are resolved". Once again, please prove that by providing a spectrum! Section 4.3, Figure 9: As the comparison of u-star shows a huge scatter, sample time/spatial series and spectra from periods when the sonic and lidar data agree and disagree, respectively could provide more insight into the differences. Have you looked at the data in more detail? Any additional information would be good? Page 15, line 8: 'the largest scales of turbulence are observed if the time window length exceeds the integral time scale ... ': Although this is correct a discussion about the error should be added, i.e. what is the error due to poor statistics if the time window is 5 or 10 min only. Page 19, line 13: examples that turbulence can significantly vary spatially is shown in Maurer et al. (2016) doi:10.5194/acp-16-1377-2016. Page 20, line 1: examples for spatial variations in the mean wind due to local flow (valley wind) is also demonstrated in Adler et al. (2014) doi: 10.1007/s10546-014-9957-8. Page 20, line 9: (spatial resolution): what is the physical resolution? See comment above. Page 21, line 28: Please rewrite the sentence "Although the sonic anemometer observations agreed most poorly with RHI-measured TKE and TI" into "Although the RHI-measured TKE and TI agreed most poorly with sonic anemometer observations" because sonic observations are considered to be the "truth".

Typing errors Figure caption Figure 2: '..... shown (c)': delete 'shown' Figure 7: 'u

integral scale' should be 'l integral scale' Page 15, line 9: '10 - –100 s' should be '10 – 100 s' Page 20, line 4: "e.g., Mann et al. . . " instead of " . . . i.e."

---

## Referee Comment (RC2) · Anonymous Referee #2 · 11 Mar 2017

The authors compare different methods for calculation of turbulence parameters from measurements with a single Doppler lidar. The topic is up-to-date and is very important for the further development of different scientific disciplines and the further technical development, e.g., of wind power plants. The authors have put considerable work into the paper and AMT is the right journal to publish this study. I recommend publication if the following major issues are addressed:

Major issues:

1.) At many points in the document, quite subjective descriptions of a correlation or a match are given. Please look into this issue. You could, e.g., quantify what you mean with "good", "bad", "show skill" or "accurate" once within the document and connect it with proper numbers. It will really upvalue the paper, if you make it more quantitative.

[Figure]

It will help to transport the message.

2.) p12 l10: -> This discussion must be mentioned earlier in the document. Is there really no way to estimate the uncertainties of these methods? I would doubt that. Understandably, analytic error calculation is extremely difficult for this kind of evaluations. However, numerical methods exist that can yield an error estimation for certain kinds of noise. You could, e.g, use Monte Carlo simulations, imposing small variations on the input data and then analyze in what range the results change. With such an approach it would be possible to discriminate between measurement errors and methodical uncertainties (e.g., incomplete overlap between the tower measurements and the lidar observations). That would greatly help the interpretation especially of Figs. 4/5 and 8/9.

This topic is also connected with P.13 Line 8: "Approximately half of these outliers are negative TKE values, which were removed as discussed earlier..." -> Are those really outliers or just noisy values that happen to be close to zero. An uncertainty estimation or a more thorrough description of the 6-beam technique would help here.

3.) p9 l18: "These erroneous echoes were removed using a discontinuity-based algorithm described by Bonin and Brewer (2016)" -> Maybe it is not so easy. Such a correction is never perfect and some artifacts always remain. The kind of signal folding you experience imposes spatially confined biases on the measured signal (spanning some range gates). Some techniques may be more susceptible to this influence than others, introducing an unknown bias into the intercomparison. E.g., the six beam technique will be affected differently by spatially confined shifts in a single beam than the RHI scans. Signal folding is also no necessity for Doppler lidar measurements. They can be avoided by reducing the pulse repetition rate, which should be mentioned. Several other questions arise and have to be discussed: (a) How is it possible to identify the folding effect unambiguously in the data? (b) What percentage of data is affected? (c) What is the remaining bias after correction?

Minor issues:

p2 l7: "good": -> As described above, please quantify...

p2 10 "the long time series of staring": -> Again, please quantify. It is actually a very good question what "long" means here. You correctly cite Lenschow et al. (1994) here, but leave the calculations to the reader. Please give a rough estimation of what "long" means in this context.

p1 l4: "trusted in situ instrumentation": -> I think I know what you mean, but please give a reference of what "trusted" means in this context. Do you mean something like "officially approved by a standardization institution"?

p1 l12: "None of the methods evaluated were able to consistently accurately measure the shear velocity" -> Please discuss what accuracy is necessary to measure shear velocity "accurately". Which maximum error is allowed for which purpose?

p6 l7: Since data were collected at 2 Hz, two samples were collected 0.5 s apart -> Please decide between mentioning "2Hz" or "0.5s".

p8 l4: SNR<-27 dB -> which definition of SNR is applied here?

p16 l1: "show skill" -> Please define "skill" together with the other descriptions.

Typos:

p1 l11: Typo: "to biased" -> "to be biased"

Table2: typo at "0.547nb"

———————————————

---

## Author Comment (AC1) · 17 May 2017

The reviewer's comments are given in green, while our response is provided in black.

**Reviewer 1**

General comments: The paper presents observations from the eXperimental Planetary boundary layer Instrumentation Assessment (XPIA). The observations were used to verify Doppler lidar turbulence profiles through comparison with sonic anemometer measurements. During a 17-days period, a single scanning Doppler lidar continuously cycled through different turbulence measurement strategies: velocity azimuth display, six-beam, and range height indicators with a vertical stare. The investigation focused on turbulence kinetic energy, turbulence intensity, and shear velocity data. For evaluation, sonic anemometer measurements at six heights on a 300-m tower were available. The paper is well written and of general interest for the lidar community (scientists and users alike). I therefore recommend the paper to be published with minor revisions.

Specific comments
Section 1: the introduction is quite general and could be more concise on the topic tackled (more focused) in this investigation.
We have rewritten the paragraphs in the Intro that review the previous work and alternative approaches to turbulence calculation, to make it more evident how they are related to the current work.

Page 5, line 2: Specify which of the methods mentioned in Lenschow et al. (2000) is used to remove noise.
We have clarified here that we are fitting a structure function of the form Eq. 32 in Lenschow et al. (2000) to the autocovariance of the residuals.

Page 5, line 17: could you make a statement concerning the time window for detrending (15, 20, 30, 60 min?) on the turbulence results. Applying a 20-min window could filter out large convective cells.
We have added a statement here clarifying that the 20-min window could filter out large convective eddies when the wind speed is small, and that the effect would be exacerbated for a shorter time window or higher-order (i.e., non-linear) filtering.
For this study, the 20-min window is naturally chosen since the 6-beam technique was used for 20-min at a time at the end of every hour, as later discussed in Sect. 3.

Page 9, line 14: '... 50-m pulse width...': does that mean that physically independent measurements are (physical resolution of the lidar) 50 m?
For practical applications, yes, physically independent measurements are spaced by 50 m (the pulse width), as long as the pulse does not intercept a hard target. This has been clarified in the paper here.

Page 12, line 5: $y = bx$: Transformation of equation (9) gives $y = x\ 10b$
We regret this error in the previous version, and have modified the equation accordingly here.

Figure 5 and 9: a zero line would be helpful.
Since the plots are on a logarithmic scale, values at and below zero are not shown on these plots as they are off the scale. Thus, no changes are made.

Page 13, line 22 ("....... may be due to the inability to capture all the scales of turbulence"): Spectra should be included in order to see which scales are not captured (to prove the statement would be good)
While we agree that showing spectra would be helpful to understand the limitations of the measurements, however it is not possible to calculate spectra from the VAD technique since the measurements from a PPI scan are over a large spatial area where the pointing angle is constantly varying. Since spectra cannot be used to see which scales are captured, the integral time scale is calculated from the sonic anemometer data to determine the characteristic

maximum eddy size (see Fig. 7 and discussion), which is used to show that the PPI scan area near the lidar is often smaller than the largest eddies to support the statement made here.

Page 15,line 18: (negative u-variance values). Is this the same effect as for TKE mentioned on page 11,line 5?

Yes, this is the same effect as mentioned for TKE earlier. We have added a statement to the manuscript to explicitly affirm this.

Page 16, line 7: "the bias becomes small as most of the turbulence scales are resolved". Once again, please prove that by providing a spectrum!

Again, while we agree that spectra would be the clearest way to support this statement, we are unable to produce power spectra using time-series analysis from the measurements using the VAD technique (see response above). That is why the analysis comparing the VAD scanning circle area to the integral scale of turbulence (typical largest eddy size) is given at the end of Sect. 4.1 to support the statement made here instead of using spectrum.

Section 4.3, Figure 9: As the comparison of u-star shows a huge scatter, sample time/spatial series and spectra from periods when the sonic and lidar data agree and disagree, respectively could provide more insight into the differences. Have you looked at the data in more detail? Any additional information would be good?

We assume that the reviewer is referring to Fig. 12, not 9, as 12 shows u-star measurements and shows the large scatter pointed out by the reviewer. Again, due to the nature of how the 6-beam and VAD methods operate (see descriptions in Sect. 2.1, 2.2), it is not possible to calculate spectra or show a time series of the u, v, or w themselves. For these reasons, it is difficult to pinpoint the reasons for why the u-star measurements are often so poor using either the 6-beam or VAD technique from the Doppler lidar.

However, since u-star is calculated based on the measurements of u'w' and v'w', the accuracy of u-star itself is reliant on the validity of u'w' and v'w' measurements. As discussed in Sect. 5, u'w' and v'w' are difficult to measure since their magnitudes are typically small ($<0.1$ m^2 s^-2) and are not measured accurately by either the 6-beam or VAD techniques (see Appendix A for quantitative analysis). Since neither of these covariance terms can be accurately measured with the 6-beam and VAD techniques, it is unsurprising that u-star, the combination of these terms, measured by the lidar shows poor correlation with u-star calculated from sonic measurements. We have added the following statement in the first paragraph of Sect. 5 to point out this relationship: 'Since the individual covariance terms do not correlate well with sonic anemometer measurements, it is unsurprising that the u* values computed from either the six-beam or VAD techniques also show little correlation with u* from the sonic anemometer ($r^2=0.14$--$0.17$).'

Page 15, line 8: 'the largest scales of turbulence are observed if the time window length exceeds the integral time scale ... ': Although this is correct a discussion about the error should be added, i.e. what is the error due to poor statistics if the time window is 5 or 10 min only.

We have added a paragraph providing a discussion of the sampling errors (i.e., statistical representativeness, related to the time window) at the beginning of Sect. 4 before any of the results are shown. By placing the discussion here before the intercomparison of turbulence measurements, the material and possibility of these errors is introduced to the reader before results are interpreted. In this discussion, we provide several sentences about the error due to poor statistics if the time window is short compared to the integral time scale.

Page 19, line 13: examples that turbulence can significantly vary spatially is shown in Maurer et al. (2016) doi:10.5194/acp-16-1377-2016.

A reference to Maurer et al. (2016) has been added here.

Page 20, line 1: examples for spatial variations in the mean wind due to local flow (valley wind) is also demonstrated in Adler et al. (2014) doi: 10.1007/s10546-014-9957-8.
A reference to Adler and Kalthoff (2014) has been added here.

Page 20, line 9: (spatial resolution):what is the physical resolution? See comment above.
See response to comment above. The physical resolution of the lidar, generally speaking, is related to the pulse width (here 50 m).

Page 21, line 28: Please rewrite the sentence "Although the sonic anemometer observations agreed most poorly with RHI-measured TKE and TI" into "Although the RHI-measured TKE and TI agreed most poorly with sonic anemometer observations" because sonic observations are considered to be the "truth".
This sentence has been rewritten as indicated.

Typing errors:
Figure caption Figure 2: '..... shown (c)': delete âAŽshown'
Corrected
Figure 7: 'u integral scale' should be 'l integral scale'
Changed simply to 'Integral scale' to prevent confusion.
Page 15, line 9: '10 - −100 s' should be '10 −100 s'
Corrected
Page 20, line 4: "e.g., Mann et al..." instead of "...i.e."
Corrected

---

## Author Comment (AC2) · 17 May 2017

We thank the reviewer for their constructive comments, both general and specific. By addressing these comments, we believe the quality of the manuscript has improved. Below, we have addressed the concerns raised by the reviewer, and have made the indicated changes to the manuscript where possible and appropriate.

Please also note the supplement to this comment:
http://www.atmos-meas-tech-discuss.net/amt-2017-35/amt-2017-35-AC2-supplement.pdf

[Figure]

**Supplement:**

The reviewer's comments are given in green, while our response is provided in black.

The authors compare different methods for calculation of turbulence parameters from measurements with a single Doppler lidar. The topic is up-to-date and is very important for the further development of different scientific disciplines and the further technical development, e.g., of wind power plants. The authors have put considerable work into the paper and AMT is the right journal to publish this study. I recommend publication if the following major issues are addressed:

Major issues:
1.) At many points in the document, quite subjective descriptions of a correlation or a match are given. Please look into this issue. You could, e.g., quantify what you mean with "good", "bad", "show skill" or "accurate" once within the document and connect it with proper numbers. It will really upvalue the paper, if you make it more quantitative. It will help to transport the message.

In several places throughout the manuscript, we have added quantitative descriptions (i.e., correlation coefficients, slopes to indicate bias) to the text where it seems appropriate, including in the abstract. However, to ensure ease of the narrative when describing these data sources in the discussion/results sections, we found it preferable to use more qualitative language to indicate whether we thought these vales represented a good or bad fit with respect to the other techniques and measurements, and to avoid redundancy of the quantitative analysis already provided in the tables and figures themselves. Instead, the appropriate figures or tables are referenced in the text. In addition, Table 2 summarizes all the quantitative statistics for a quick look up by the reader. To further strengthen the quantitative analysis, we have also provided the RMSE values in the updated Table 2.

2.) p12 l10: -> This discussion must be mentioned earlier in the document. Is there really no way to estimate the uncertainties of these methods? I would doubt that. Understandably, analytic error calculation is extremely difficult for this kind of evaluations. However, numerical methods exist that can yield an error estimation for certain kinds of noise. You could, e.g, use Monte Carlo simulations, imposing small variations on the input data and then analyze in what range the results change. With such an approach it would be possible to discriminate between measurement errors and methodical uncertainties (e.g., incomplete overlap between the tower measurements and the lidar observations). That would greatly help the interpretation especially of Figs. 4/5 and 8/9. This topic is also connected with P.13 Line 8: "Approximately half of these outliers are negative TKE values, which were removed as discussed earlier..." -> Are those really outliers or just noisy values that happen to be close to zero. An uncertainty estimation or a more thorough description of the 6-beam technique would help here.

We agree with the reviewer that quantifying the uncertainty of the measurements is ideal for these intercomparisons and interpreting the results. Unfortunately, determining the magnitude of the uncertainty itself is not trivial. Entire studies have been devoted to addressing and quantifying sampling errors of turbulence and flux measurements from time-series analysis alone (see references in new discussion in paper). Since the measurement techniques presented here have are made using Doppler lidar observations over both time and space, no method has been developed yet to quantify errors with these techniques. Due to the intensive analysis required to properly attribute sampling errors from even one technique (as evidenced by multiple studies now referenced in the paper that evaluate sampling errors from time series analysis), entire studies could be done to determine the proper sampling errors for each of the techniques here separately. As such, properly quantifying and attributing sampling errors are out of the scope of this paper and is a topic for further studies. In lieu of quantifying these errors, we have added a more thorough discussion of sampling errors earlier in the paper before the results are presented (last paragraph before Sect. 4.1).

We do note that the input data (line-of-sight velocity measurements) are already contaminated with random error (i.e., noise). This is covered in Sect. 2 (see Eq. 1 & 2). However, using

established techniques using the autocovariance of the line-of-sight velocity measurements, we have been able to quantify and remove this noise from all of the scans except for the RHIs (as is discussed in Sect. 2). As such, random errors are not anticipated to be a significant source of error except for the RHIs, as discussed in the manuscript. With this approach, we feel that a Monte Carlo simulation is not appropriate to assign errors, as these noise effects are already quantified and removed.

The negative TKE values are simply outliers based on the definition used in the manuscript, being more than 1 order of magnitude difference between sonic anemometer and observations (see p. 13, line 20). As mentioned above, quantifying the sampling error of six-beam measurements is difficult due to the spatio-temporal nature of the technique and out of the scope of the paper. However, we do believe the negative values are thought to be related to the sampling error of the measurement (since there is no other plausible explanation), and state this where the negative values are first discussed on p.12 line 2.

3.) p9 l18: "These erroneous echoes were removed using a discontinuity-based algorithm described by Bonin and Brewer (2016)" -> Maybe it is not so easy. Such a correction is never perfect and some artifacts always remain. The kind of signal folding you experience imposes spatially confined biases on the measured signal (spanning some range gates). Some techniques may be more susceptible to this influence than others, introducing an unknown bias into the intercomparison. E.g., the six beam technique will be affected differently by spatially confined shifts in a single beam than the RHI scans. Signal folding is also no necessity for Doppler lidar measurements. They can be avoided by reducing the pulse repetition rate, which should be mentioned. Several other questions arise and have to be discussed: (a) How is it possible to identify the folding effect unambiguously in the data? (b) What percentage of data is affected? (c) What is the remaining bias after correction?

We agree with the reviewer that the cited technique to remove range folded echoes is not perfect, largely since it relies on contextual information. This technique works better on RHI and PPI scans than it does on measurements at single beam positions, due to the necessity of having data nearby spatially and temporally. While showing and discussing the results of the intercomparison, we specifically point out in the manuscript that the large number of outliers in the six-beam TKE measurements are largely a consequence of range folded echoes not being removed due to short amount of time at each beam position (see p. 13, lines 22-28).

We acknowledge that range folding does not appear in all Doppler lidar measurements. However, the commercial Doppler lidar systems that have been increasingly used by a diverse set of users in the past 5 years (Leopshere and Halo systems) have PRFs ~10-20 kHz and have been documented to be susceptible to range folded echoes (see Päschke et al. (2015) and Bonin and Brewer (2017)). By clearly stating (see p. 9, line 17) that the high PRF (20 kHz) of the system is why range folding is an issue, it is implied that range folding is not an issue for low PRF systems. Thus, we do not see a need to expand upon the issue in the manuscript.

While these anomalous echoes do affect the data quality and we feel that they are important to mention, they are not the main focus of the manuscript and much of the data (as determined by manual inspection) is not affected after the described and referenced QC methods are applied. Going into details about these echoes, their characteristics, frequency of occurrence, etc. requires rigorous analysis itself and is outside of the scope of the paper. Instead, we have added a statement in the manuscript to refer the reader to Bonin and Brewer (2017) for these characteristics and how these anomalous echoes are identified.

We have provided a statement in the manuscript indicating that 5.6% of data points were flagged and removed, which we believe to be an overestimate (since we conservatively remove

suspect data points). To really address this issue, a low-PRF system, such as our HRDL, needs to be synchronized with the high-PRF system scanning the same positions at the same time, to accurately quantify the percentage of data from that high-PRF system that is range folded through a direct intercomparison of measurements along each beam. This was not done during XPIA, and to our knowledge has not been done any other time before. Thus, the exact percentage of data affected by range folding is not known, until we can perform such as experiment.

Minor issues:
p2 l7: "good": -> As described above, please quantify...
We have decided to simply remove the word 'good', and provide a reference for the reader if they are interested in the details of the results of the cited study.

p2 10 "the long time series of staring": -> Again, please quantify. It is actually a very good question what "long" means here. You correctly cite Lenschow et al. (1994) here, but leave the calculations to the reader. Please give a rough estimation of what "long" means in this context.
As we have rewritten much of the introduction, there is no obvious location to discuss this here. However, we have added a paragraph in the beginning of Sect. 4 discussing sampling errors, in which we give an estimation of how long a time series needs to be under convective conditions (1-2 hours) and stable conditions (5-10 min).

p1 l4: "trusted in situ instrumentation": -> I think I know what you mean, but please give a reference of what "trusted" means in this context. Do you mean something like "officially approved by a standardization institution"?
We have clarified in the introduction at p. 3 l 15 that sonic anemometers are a commonly used reference device, and have provided a reference to International Energy Agency (IEA) report that supports this claim. We have decided to keep the wording unchanged in the abstract (at p1 l4), as we feel it is not the appropriate location to give a reference for this.

p1 l12: "None of the methods evaluated were able to consistently accurately measure the shear velocity" -> Please discuss what accuracy is necessary to measure shear velocity "accurately". Which maximum error is allowed for which purpose?
As the reviewer states, the maximum error allowed for a measurement to be useful depends on the purpose or use of the measurement.  We do provide a general discussion on how accurate measurements need to be depending on the application in the paragraph starting on p. 2 l. 33. But we do not believe that the essence of this study is to sort out whether a measurement is accurate enough for a given purpose. That is for users to decide based on their specific objectives.  Instead, the objective here is to analyze the measurement techniques to determine systematic biases and if measurements are of any use at all to any audience.  In the particular sentence referred to here, we state that stress velocity was not found to be accurate by any of the methods tried. This is true regardless of the use of the measurement as the correlation coefficient between the sonic and VAD or six-beam u-star was 0.171 and 0.147 respectively and the scatter was large across all values, indicating very little correlation, and that the lidar measured u-star was insufficiently accurate for any purpose.

p6 l7: Since data were collected at 2 Hz, two samples were collected 0.5 s apart -> Please decide between mentioning "2Hz" or "0.5s".
We have removed the 2 Hz statement and now are using only 0.5 s here.

p8 l4: SNR<-27 dB -> which definition of SNR is applied here?

A statement has been added here to clarify that the SNR values were taken as the carrier-to-noise ratio produced by the lidar manufacturer's processing algorithms. Since the processing algorithms are proprietary, we are unsure about the details of how the SNR/CNR is exactly calculated and the exact definition used. However, the values of SNR/CNR should be comparable for measurements from other Leosphere Doppler lidars.

p16 l1: "show skill" -> Please define "skill" together with the other descriptions.
We have added a definition of 'skill' here as showing correlation between the lidar and the sonic anemometer (reference instrument).

Typos:
p1 l11: Typo: "to biased" -> "to be biased"
Corrected
Table2: typo at "0.547nb"
Corrected